# Open Iliac Conduits Enabling the New Era of Endovascular Aortic Repair in Hostile Iliofemoral Anatomy: A Single-Center Retrospective Study

**DOI:** 10.3390/medicina62010017

**Published:** 2025-12-22

**Authors:** Konstantinos Litinas, Michalis Pesmatzoglou, Nikolaos Kontopodis, Ioannis Kakisis, Christos V. Ioannou

**Affiliations:** 1Vascular Surgery Unit, Department of Vascular Surgery, School of Medicine, University of Crete, 71100 Heraklion, Greece; litinask@gmail.com (K.L.); michalis.pesmatzoglou@gmail.com (M.P.); ioannou@med.uoc.gr (C.V.I.); 22nd Department of Vascular Surgery, Attikon University Hospital, 15772 Athens, Greece; kakisis@med.uoa.gr

**Keywords:** aortic aneurysm, TEVAR, BEVAR, FEVAR, iliac conduit, endovascular repair

## Abstract

*Background and Objectives*: Hostile iliofemoral anatomy (HIA) challenges large-bore access in thoracic, branched, or fenestrated endovascular aortic repair (t/b/fEVAR). Retroperitoneal open iliac conduit (ROIC) enables safe delivery, but data in complex t/b/fEVAR are scarce. *Materials and Methods*: This retrospective single-center cohort study (2017–2025) of 80 t/b/fEVAR patients followed STROBE guidelines. Eight (10%) required elective ROIC for HIA (small iliac diameter < 7 mm or occlusive disease). Outcomes were compared to 23 no-conduit complex endovascular aortic repair cases. *Results*: ROIC patients [50% female, 87.5% smokers] had higher PAD [62.5% vs. 17.4%, *p*-value = 0.015]. All ROICs were elective [vs. 69.5% no-conduit, *p*-value = 0.076]; indications: Type V TAAA [50%], synchronous aneurysms (25%), Type II TAAA [12.5%] and arch aneurysm [12.5%]. Median operative time [365 vs. 200 min, *p*-value = 0.002], blood loss [1190 vs. 600 cc, *p*-value < 0.001], and contrast [420 vs. 300 cc, *p*-value = 0.004] were higher. Technical success was 100% [8/8] vs. 86.9% [20/23] (*p*-value = 0.28), and clinical success was 87.5% vs. 78.2% (*p*-value = 0.569). Median ICU stay [3 d vs. 2 d, *p*-value = 0.817] and hospital stay [12 d vs. 9 d, *p*-value = 0.404] were prolonged, albeit without statistically significant differences. In-hospital mortality was similar (12.5% vs. 17.4%, *p*-value = 0.746) between groups. One ROIC patient had intraoperative cardiac arrest [sheath dislodgement]; another required a covered stent for anastomotic rupture. At 12-month follow-up, one Type III endoleak required relining. *Conclusions*: Planned retroperitoneal open iliac conduits achieved 100% technical success in patients with hostile iliofemoral anatomy, without ischemic complications, despite longer operative times and higher blood loss. ROIC remains a safe and indispensable technique that extends complex endovascular aortic repair to otherwise ineligible patients.

## 1. Introduction

Endovascular repair of thoracic, thoracoabdominal, and complex abdominal aortic pathology via thoracic, branched, or fenestrated endografts [t/b/fEVAR] has become the preferred modality due to reduced early morbidity versus open surgery [1,2]. However, device delivery mandates adequate iliofemoral access. Instructions for Use [IFU] restrict deployment when the external iliac diameter < 7 mm, the circumferential calcification > 50%, there is occlusive disease, the acute iliac angulation > 90°, or prior iliofemoral grafts exist—collectively termed hostile iliofemoral anatomy [HIA] [3,4].

Management options include direct iliac exposure, balloon angioplasty, “pave-and-crack” techniques, endoconduits [EC], or retroperitoneal open iliac conduits [ROIC] [4,5]. Historically, ROIC is considered the gold standard for severe HIA [5]. Endoconduits offer a minimally invasive alternative with low early mortality [6], yet they risk gluteal necrosis, buttock claudication, or spinal cord ischemia [SCI]—particularly in extensive aortic coverage or unstaged emergency repairs requiring internal iliac occlusion [6].

HIA prevalence reaches 9–21% in TEVAR and up to 64% in f/bEVAR [5,7,8]. Conduit use [open or endovascular] averages 17% across 16,855 EVAR cases [9]. ROIC extends the operative time [~67 min] and transfusion risk [5,7] but provides controlled hemostasis and durable access. According to previous research, retroperitoneal iliac exposure during endovascular repair of complex aortic aneurysms has been associated with lower technical success rates, albeit without increasing intraoperative access complications or major adverse events [7]. Specifically, an 28% major adverse events rate was recorded in the conduit cohort, with no statistically significant difference from the non-conduit, while the technical success rate was 83% among patients in the conduit compared with 98% in the standard group [7]. Others summarized available evidence conducting a systematic review and reported a pooled estimate for overall technical success > 95%, on top of a non-negligible rate of bleeding (10%), wound (5%) and any peri-procedural complications (32%), among patients requiring an iliac conduit [9]. Notably, this metanalysis reported a statistically significant almost three-fold odds ratio for death and a 2.38 odds ratio for bleeding complications among patients requiring a conduit. Contemporary series rarely isolate ROIC outcomes in elective t/b/fEVAR, nor do they compare them to no-conduit cohorts under modern reporting standards, and comparative analyses are still needed to determine the best strategy to address challenging iliac artery accesses.

We present a STROBE-compliant retrospective analysis of all t/b/fEVAR procedures (2017–2025) at a single center, focusing on patients requiring planned ROIC for HIA. We compare perioperative metrics, complications, and 12-month reinterventions to no-conduit controls, emphasizing ROIC’s role. Our objective is to define ROIC’s contemporary safety, efficacy, and indispensable niche in the endoconduit era.

## 2. Materials and Methods

### 2.1. Study Design

We conducted a retrospective observational study of all patients treated with t/b/fEVAR in our department between January 2017 and November 2025, according to the STROBE statement for observational studies [10].

Specifically, we identified all of the t/b/fEVAR patients, who underwent endovascular aorta repair deploying large-bore sheaths > 22 Fr, thus requiring open access [cut-down].

Our study population included patients who required a retroperitoneal open iliac conduit [ROIC] to secure iliac access to complete the endovascular procedure. All patients successfully completed a twelve-month follow-up.

### 2.2. Inclusion Criteria

All patients who underwent thoracic or complex abdominal aorta endovascular reconstruction [TEVAR, BEVAR or FEVAR] for thoracic/thoracoabdominal pathology were included in our study. This included patients with acute thoracic aortic syndrome necessitating treatment [aortic dissection—AD, intramural hematoma—IMH, penetrating aortic ulcer—PAU, and thoracic aorta rupture], patients with thoracic, thoracoabdominal or complex abdominal aorta aneurysm or pseudoaneurysm, and patients with acute thoracic aortic injury after trauma.

After the initial screening, patients with a hostile iliac artery requiring an iliac conduit to secure distal access were identified. Those patients were then compared to others who did not require ROIC placement.

### 2.3. Exclusion Criteria

Patients who underwent standard EVAR, those who had percutaneous access (typically employed with a delivery profile < 21 fr), as well as patients with aortic pathology who were managed conservatively, were excluded from this case series.

### 2.4. Endpoints

Information collected included patients’ demographic characteristics [age, gender, and comorbidities], the modality of treatment, the type of repair [including type of endograft deployed], and intraoperative [type of anesthesia, blood loss, IV contrast used, operative time] and postoperative details.

The primary endpoints included the technical and clinical success rates and in-hospital mortality. The secondary endpoints included intraoperative information such as the procedural time, contrast volume, blood loss, postoperative information such as complications, the need for re-intervention, the need for and length of stay in the ICU, the length of hospital stay, and finally, the follow up outcomes.

Technical success was assessed as defined per the Oderich reporting standards, as the successful introduction and deployment of the endograft with/without ROIC, without type I/III endoleak on completion angiography, and no open conversion or death within 24 h [11].

Clinical success was also assessed per the Oderich definition of a successful endograft deployment, without aneurysm-related death, type I/III endoleak, graft infection or thrombosis, further aneurysm expansion or rupture, or need for open conversion [11].

## 3. Results

### 3.1. Study Population

After the initial search in our database, 80 patients treated with t/b/fEVAR were identified. Among the patients initially identified, a retroperitoneal iliac conduit was placed in eight patients, and those patients composed our target group.

However, after assessing the patients, we identified that all patients who had a conduit placed were treated with complex endovascular procedures, while the majority of the no-conduit group underwent standard TEVAR [*n* = 49]. Only 23 of those patients underwent complex endovascular aorta repair. Thus, to ensure comparable severity levels of interventions, we decided to exclude all of the standard TEVAR patients from the statistical analysis. Therefore, the statistical analysis included only patients undergoing fenestrated and branched EVAR with or without conduits.

The flowchart presenting the abovementioned results is depicted in Figure 1.

### 3.2. Patient and Procedural Characteristics

The patients’ characteristics that were analyzed included smoking and the previous medical history, regarding arterial hypertension [AH], hyperlipidemia [HLP], diabetes mellitus [DM], cardiovascular disease [CVD], and peripheral arterial disease [PAD]. The analysis of the patients showed no difference between genders in the iliac-conduit group [male: *n* = 4 and female: *n* = 4]; however, in the no-conduit group, all patients were male, with the difference being statistically significant. Almost all of the patients in the conduit group were active smokers [*n* = 7], with a history of AH [*n* = 8], HLP, CVD, PAD, and COPD [*n* = 5]. Comparing the two groups [conduit versus no-conduit], more patients had a history of peripheral artery disease in the conduit group [62.5% versus 17.3%], attributed most probably to the severely stenotic/calcified iliofemoral axis, achieving statistical significance. Most of those patients had symptoms of intermittent claudication [*n* = 5], and only two patients had symptoms of rest pain [=critical limb ischemia], which in both cases subsided after restoring the iliofemoral pathology. Patients’ comorbidities are illustrated in detail in Table 1.

All of the patients where a conduit was placed were treated under elective settings, due to Type V thoracoabdominal aneurysm [*n* = 4], Type II TAAA [*n* = 1], aortic arch aneurysm [*n* = 1], or synchronous descending thoracic and infrarenal abdominal aortic aneurysm [*n* = 2]. The patients underwent complex procedures including TEVAR and BEVAR [*n* = 2], TEVAR and EVAR [*n* = 1], TEVAR, BEVAR, and EVAR [*n* = 2], BEVAR and EVAR [*n* = 1], TEVAR and FEVAR [*n* = 1], or arch debranching [T-arch EVAR] [*n* = 1]. None of the patients had their left subclavian artery [LSA] covered. Further, none of the patients had their operations staged, meaning placing the iliac conduit at first and after a period of time reconstructing the aorta pathology.

Most of the patients in the no-conduit group were also operated under elective settings [69.5%], mainly due to pararenal aneurysm [*n* = 9], Type V TAAA [*n* = 6], Type IA endoleak in patients previously operated with endovascular aneurysm repair [*n* = 3], Type IV TAAA [*n* = 1], suprarenal aneurysm [*n* = 1], TAAA Type III with synchronous AAA [*n* = 1], ATBAD with TAAA formation [*n* = 1], and CTBAD with TAAA formation [*n* = 1]. The abovementioned results are stated in Table 2.

Among the patients where an iliac conduit was placed, this was mandated due to the small diameter of the iliac artery < 7 mm [*n* = 3] or due to excessive iliac artery stenosis-occlusion [*n* = 5]. The iliac conduit was placed using an open retroperitoneal approach, following the usual approach [*n* = 8]. Specifically, a standard lower abdominal quadrant oblique incision was made, deepened through the subcutaneous tissue and through the external and internal oblique and transversalis abdominal muscles, reaching the preperitoneal and subsequently the retroperitoneal space. After exposing and controlling the iliac vessels, the conduit was established. The iliac conduit was placed either in the common iliac artery [*n* = 7] or in the aortic bifurcation [*n* = 1] (Figure 2), in a case of excessively narrow common iliac arteries [diameter around 2 mm]. In 87.5% of the patients [*n* = 7], a Dacron graft was used, while an ePTFE ringed graft was utilized in one patient. After completion of the procedure, conduit ligation was conducted in four patients [50%], with a two-line meander-like suture line, while in two patients, iliofemoral bypass was utilized, in one patient, an aortoiliac bypass was constructed, and finally, in one patient, the conduit was left inside [with its distal end ligated] to further utilize it for a second procedure. Three patients required further conduit placement in the right upper limb, either in the axillary artery [*n* = 2] or in the subclavian artery [*n* = 1]. General anesthesia was utilized in all of the patients in the conduit group and in 95.6% of the patients in the no-conduit group [*n* = 22], with one patient being operated under spinal [4.3%] anesthesia.

### 3.3. Primary Outcomes

#### 3.3.1. Technical Success

Technical success was achieved in all patients in the conduit group and in 20/23 patients in the non-conduit group, as per Oderich reporting standards. This difference was not statistically significant (*p* = 0.28). Three patients in the non-conduit group had intraoperative rupture of the aneurysm, requiring massive blood transfusion, resulting in complications such as disseminated intravascular coagulation (DIC), transfusion-related acute lung injury (TRALI), and finally, death within 24 h of the operation.

#### 3.3.2. Clinical Success

Clinical success was observed in all but one patient in the conduit group and in 18/23 patients in the non-conduit group, without evidence of a statistically significant difference (*p* = 0.569). One patient in the conduit group suffered from iliac limb thrombosis leading to non-reversible ischemia, thus resulting in an above-the-knee amputation. Among the patients in the non-conduit group, three of them did not achieve technical success, thus excluding them by definition from achieving clinical success. For the remaining two patients, one had the branch to the left renal artery thrombosed, and the second one had a type IB endoleak in the 30 d follow-up CTA, necessitating reintervention in both cases, either with AngioJet thrombectomy and renal artery stent relining or with iliac limb stent relining, respectively.

#### 3.3.3. In-Hospital Mortality

In-hospital mortality occurred in 1/8 patients in the conduit and in 4/23 patients in the non-conduit group. The patient in the conduit group deceased during her 100th post-operative day, after a burdensome hospitalization complicated with iliac limb thrombosis requiring femoral amputation, intestinal ischemia requiring multiple enterectomy operations from general surgeons, two episodes of cardiac arrest, and finally an upper respiratory infection, which finally resulted in the patient’s death. In the non-conduit group, three patients died within 24 h of the operation, as previously stated in Section 3.3.1, while one patient died during his 83rd post-operative day, after a prolonged ICU stay, due to severe recurring upper respiratory infection with pandrug-resistant (PDR) bacteria.

### 3.4. Secondary Outcomes

#### 3.4.1. Intraoperative Details

The total operative time (median) was 365 min in patients where ROIC was placed, in contrast to 200 min in patients when ROIC was not placed. The median total blood loss [TBL] was 1190 cc versus 600 cc, and the median intravenous contrast used was 420 cc versus 300 cc, respectively. The establishment of an iliac conduit increased the total operative time, the intraoperative blood loss, and the intravenous administered contrast, achieving statistical significance. The increased blood losses can partially be attributed to the open approach, as well as to the mismatch between the iliac conduit and the endograft delivery system, with inefficient hemostasis around the introducer sheath. Additionally, in one case, an accidental removal of the sheath from the iliac conduit caused severe hemorrhage, with the patient suffering intraoperative cardiac arrest due to hemorrhagic shock. In another case, rupture of the anastomosis occurred during sheath removal after graft deployment, resulting in severe blood loss, which was managed by placing a covered stent. Noteworthy, rupture of the iliac artery is a known complication of ROIC, with a rate around 2–5%, typically during sheath withdrawal in calcified vessels [5,7]. The results regarding the technique during placement of the conduit are summarized in Table 3, while the intraoperative details are depicted in Table 4.

#### 3.4.2. Postoperative Outcomes

All of the patients treated with ROIC were transferred to the ICU immediately after the operation, with a median ICU stay of 3 days, in contrast to the patients without an iliac conduit, where only 47.8% [*n* = 11] of them were transferred to the ICU, for a median stay of 2 days.

The post-operative complications were notable both in the conduit and the no-conduit groups, with an overall complication rate of 62.5% [*n* = 5] and 78.3% [*n* = 18] respectively, which included endoleak in 25% [*n* = 2] and 21.7% [*n* = 5], neurological complications in 25% [*n* = 2, with Horner syndrome development in one patient] and 17.4% [*n* = 4], respiratory complications including pulmonary embolism and pulmonary infection in 50% [*n* = 4] and 13% [*n* = 3], bleeding complications necessitating re-intervention, aneurysm rupture in 12.5% [*n* = 1] and 12.5% [*n* = 1], and acute kidney injury in 12.5% [*n* = 1] and 21.7% [*n* = 5], respectively. Additionally, two patients from the no-conduit group required hemodialysis during their hospitalization. Regarding perioperative endoleak rates, more prevalent was Type II-EL [*n* = 5], followed by Type IB-EL [*n* = 1] and Type-III-EL [*n* = 1]. All Type I/III-EL were subjected to a new operation, whereas type II-EL were managed conservatively. One patient from the conduit groups had Type IA-EL, which was assessed intraoperatively with proximal balloon angioplasty using the aorta molding balloon with recession of the endoleak.

The mean hospital stay was 12 days for the conduit group versus 9 days for the no-conduit group, with an intra-hospital mortality calculated around 12.5% [*n* = 1/8] and 17.3% [*n* = 4/23], respectively. Although patients with an iliac conduit established had both a prolonged ICU and hospital stay, their intrahospital mortality rates were lower compared to the control group. This could be attributed probably due to the small sample of patients treated with ROIC.

Different outcomes between male and female patients in the conduit group were not noted.

Post-operative complications, as well as the need for and duration of ICU and the total hospital stay are stated in Table 5.

#### 3.4.3. Follow-Up and Need-for-Reintervention

Patients were followed up with CT angiography at 1, 6, and 12 months and annually afterwards, considering that no pathological findings were identified. In cases of endoleak or sac enlargement, more frequent imaging, at 6-month intervals, was applied. All of the patients that were discharged alive from the hospital have successfully completed 1-month, 6-months, and 12-month follow-ups, while only one has completed the 36-month follow-up, and only two have completed the 18-month follow-up.

At the 1-month follow-up CTA, a Type III endoleak was identified in a patient who underwent BEVAR and who was treated with relining of the susceptible stent–graft. In another patient, a large retroperitoneal hematoma was identified, without evidence of active extravasation, rupture, or pseudoaneurysm formation and was thus treated conservatively. At the 6-month follow-up, a type II EL was identified in a patient, which was treated conservatively. However, during the follow-up, he had an episode of SMA thrombosis, which led to intestinal ischemia, and finally, he died. No further complications were identified in the 12-month follow-up.

## 4. Discussion

In the literature, female gender, as well as a history of smoking, COPD, and PAD, have been associated with a higher rate of iliac conduit utilization, mostly because of the smaller vessels commonly found in female patients [7,9]. These findings are consistent with the results of our study, where all of the patients in the no-conduit group were male, while all female patients that were analyzed required an iliac conduit placement.

Hostile iliofemoral anatomy was first described in 2002, as “iliac morphology score [IMS]”, which assessed the length and extent of calcification or occlusion, the vessel diameter, and the tortuosity iliac artery index [6]. The updated hostile iliac artery criteria include the extent of calcification > 50% of the vessel–lumen or the presence of occlusive disease, an acute iliac angle > 90 degrees, an external iliac artery diameter < 7 mm, or the presence of previously placed aortoiliac or femoral surgical-placed or endovascular grafts [4,8]. Hostile iliofemoral anatomy [HIA], requiring further management with placement of ROIC, has been calculated to be around 2–18% of all EVAR cases and 9–21% of all TEVAR cases [5,7]. Consequently, in a retrospective cohort study of Galitto et al., the incidence of HIA in patients undergoing f/bEVAR was calculated around 64% [*n* = 60]; among them, further procedures to ensure endograft or contralateral sheath access placement were needed in 32 patients, with surgical conduit in 14 patients [8]. In our case series, HIA was identified in 17 patients [21.2% of the total population], with a further need for ROIC placement in eight patients [10% of the total population]. In the rest of the patients, the HIA barrier was surpassed with intraoperative angioplasty of the iliac artery, with/without stent placement. No endoconduits were deployed, mainly due to the lack of equivalent experience in our department. A meta-analysis performed by Giannopoulos et al. calculated iliac conduit placement [open or endovascular] in 17% of 16.855 patients treated with endovascular aneurysm repair, without clarifying between HIA-friendly iliac anatomies [FIA] [9].

In the literature, the use of conduits was associated a with prolonged operative time, by 67 min, as well as a higher risk for intraoperative transfusion [5,7]. This is consistent with our results. In a comparative study by Motta et al., the estimated blood loss [EBL] after cAAA treatment was calculated around 431 ± 457 cc [12], which is lower than that calculated in the conduit group. The increased intraoperative blood loss and, consequently, the need for further transfusion could be explained due to the requirement for retroperitoneal access, with consequent blood loss, as well as due to inefficient hemostasis around the delivery system through the conduit. Suture-line disruption leading to blood loss can and does happen on occasion while passing large rigid devices through relatively narrow anastomotic openings, especially when the common iliac artery is small and/or diseased and thin-walled [13,14].

In addition, patients treated with conduits received higher doses of intravenous contrast compared to the no-conduits group.

Overall, the technical success [TS] in our case series in the group where the iliac conduit was placed was 100%, which is consistent with the published results, which is around 94% [9] and 96% in cases of f/bEVAR [8].

In the published literature, the use of iliac conduits has been associated with an increased risk for perioperative complications [including cardiac, pulmonary, and bleeding complications] and perioperative mortality rates, compared to no conduits [5,7,9,15], with many authors calculating the perioperative complication rates around 32% and perioperative mortality around 12% in cases of conduit use, while others state that those complex procedures [f/bEVAR and TEVAR] carry by themselves high rates of perioperative morbidity, around 40% and 28–41.9%, respectively [4,7,9]. However, in our case series, while post-operative complications were notable both in the conduit and no-conduit group, higher rates were identified in the second group [62.5% versus 78.3%, respectively], including higher endoleak rates, respiratory infection, and neurological complications. Hence, one could suggest that the urgency and not the use of conduits themselves could drive early morbidity after f/bEVAR.

The 30 d mortality was calculated at 0% for the conduit group and 13% for the no-conduit group, while the literature averages the mortality around 12% [9]. Noteworthy, the rest of the intrahospital deaths [one patient that died in the conduit group and another one in the no-conduit group] were at their 100th and 85th hospital day, respectively, due to resistant respiratory infection and bacteremia, with no deaths directly related to the use of a conduit.

In our department, endoconduits were not used during this study period, primarily due to our limited experience with this technique and concerns regarding the potential ischemic complications associated with internal iliac artery coverage in patients undergoing extensive aortic repairs.

Limitations of the current study include the small sample size, the retrospective design, lack of endoconduit comparison, and the absence of hybrid OR, with synchronous reliance on angio-suite conversion. Regarding the small sample size, one could argue against performing a formal statistical comparison between these small patient groups. We chose to perform a statistical comparison, recognizing the high potential for a Type I and II statistical error, instead of just providing a descriptive report of the outcomes. Moreover, taking into account that the focus of the current report is to provide contemporary results of complex EVAR with the use of open surgical conduits to overcome narrow access, rather than introducing a new technique, the novelty of the present findings may be modest. Additionally, the loss of long-term follow-up is apparent. Of note, many of the patients treated in our department are either from rural areas or from abroad, setting a physical barrier for successful and on-time follow-up. Future directions could include standardized ROIC protocol, regarding the graft sizing, ligation versus bypass, cost-effectiveness studies, and multicenter registries. Finaly, the current study did not consider patients undergoing open surgical repair of complex aortic aneurysms. Patients selected for fenestrated and/or branched endovascular repair via open iliac conduits represent a distinct group from those selected for open surgery, typically presenting more extensive aortic disease and comorbidity profiles that make them poor candidates for open surgery. Accordingly, introducing a comparison with open repair did not seem methodologically appropriate, taking into account that these groups are inherently non-comparable and subject to substantial selection bias. For this reason, the current study focuses on the role and outcomes of open iliac conduits within the context of endovascular management, where they serve as an adjunct to facilitate implantation of endografts in patients with hostile iliac anatomy.

In the modern era of endovascular aortic repair, the widespread adoption of fenestrated, branched, thoracic, and arch endografts has dramatically expanded the proportion of patients eligible for the minimally invasive treatment of complex thoracic, thoracoabdominal, and arch aneurysms. However, these technological advances have simultaneously increased the demand for reliable large-bore iliofemoral access, exposing the limitations in patients with truly hostile iliac anatomy. New generation endografts that would be delivered through smaller profile devices will probably become available in the future. This may allow for an easier and safer deployment of endografts through narrow access vessels, but while such progress is still awaited, iliac conduits remain an option in the armamentarium of vascular surgeons.

## 5. Conclusions

Our single-center experience confirms that retroperitoneal open iliac conduits remain an indispensable tool in contemporary practice. When severe calcification, occlusive disease, extreme tortuosity, or small-caliber external iliac arteries preclude safe transfemoral delivery of next-generation endografts, the deliberate use of a surgical conduit converts anatomical exclusion criteria into technical feasibility.

Although placement of an open conduit is associated with longer operative times, higher blood loss, and more intensive postoperative care, it offers a controlled durable access solution that enables the full spectrum of modern branched, fenestrated, and total-arch endovascular reconstructions.

In total, planned retroperitoneal open iliac conduits are a safe and useful enabling technique for complex endovascular aortic repair in patients with hostile iliofemoral anatomy, achieving high technical and clinical success without added early ischemic complications, despite longer and more resource-intensive procedures.

## Figures and Tables

**Figure 1 medicina-62-00017-f001:**
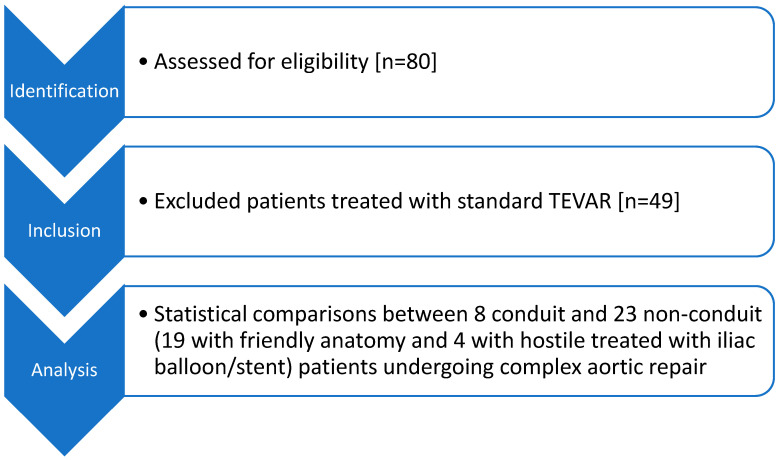
Flowchart of the study protocol/selection process.

**Figure 2 medicina-62-00017-f002:**
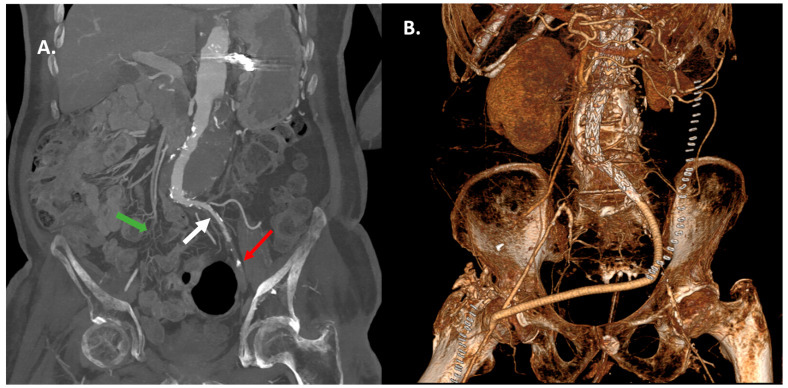
A representative case with challenging access is presented. (**A**) Preoperative multiplanar reconstruction of a patient with a thoracoabdominal aortic aneurysm and an occluded right common iliac artery (green arrow), narrow left common iliac (while arrow), and occluded left external iliac artery (red arrow). The patient had previously been subjected in a left-sided above-knee amputation. (**B**) Post-operative 3D reconstruction indicating the placement of a conduit, which was anastomosed in the right femoral artery after insertion of the endograft.

**Table 1 medicina-62-00017-t001:** Summary of patient characteristics.

Patient Group/Parameter	Total [*n* = 31]	Conduit [*n* = 8]	No-Conduit [*n* = 23]	*p*-Value
Age	71	70	71	0.760
Gender	Male: *n* = 27 Female: *n* = 4	Male: *n* = 4 [50%] Female: *n* = 4 [50%]	Male: *n* = 23 [100%] Female: *n* = 0 [0%]	<0.001
Smoker	N = 22	N = 7 [87.5%]	N = 15 [65.2%]	0.232
Arterial Hypertension	N = 26	N = 8 [100%]	N = 18 [78.2%]	0.150
Hyperlipidemia	N = 18	N = 5 [62.5%]	N = 13 [56.5%]	0.761
Cardiovascular Disease	N = 18	N = 5 [62.5%]	N = 13 [42.5%]	0.760
Peripheral Artery Disease	N = 9	N = 5 [62.5%]	N = 4 [17.3%]	0.015
Diabetes Mellitus	N = 8	N = 4 [50%]	N = 4 [17.3%]	0.069
Chronic Obstructive Pulmonary Disease	N = 15	N = 5 [62.5%]	N = 10 [43.4%]	0.353
Chronic Kidney Disease	N = 9	N = 1 [12.5%]	N = 8 [34.7%]	0.232

**Table 2 medicina-62-00017-t002:** Anatomic details and type of treatment.

Patient Group/Parameter	Total [*n* = 31]	Conduit [*n* = 8]	No Conduit [*n* = 23]	*p*-Value
Mode of admission	Emergency: *n* = 7 Elective: *n* = 24	Emergency: *n* = 0 Elective: *n* = 8 [100%]	Emergency: *n* = 7 [30.4%] Elective: *n* = 16 [69.5%]	0.076
Type of aneurysm	Pararenal aneurysm: *n* = 9 Type II TAAA: *n* = 1 Arch aneurysm: *n* = 1 Type V TAAA: *n* = 10 Type IA-EL after previous EVAR procedure: *n* = 3 Type IV TAAA: *n* = 1 Suprarenal aneurysm: *n* = 1 Type III TAAA: *n* = 1 TBAD with TAAA formation: *n* = 1 ATBAD with TAAA formation: *n* = 1 Synchronous DTAA and Infrarenal AAA: *n* = 2	TAAA type V: *n* = 4 [50%] TAAA type II: *n* = 1 [12.5%] Arch: *n* = 1 [12.5%] Synchronous DTAA and Infrarenal AAA: *n* = 2 [25%]	Pararenal aneurysm: *n* = 9 [39.3%] Type V TAAA: *n* = 6 [26.2%] Type IA-EL after previous EVAR procedure: *n* = 3 [13%] Type IV TAAA: *n* = 1 [4.3%] Suprarenal aneurysm: *n* = 1 [4.3%] Type III TAAA: *n* = 1 [4.3%] TBAD with TAAA formation: *n* = 1 [4.3%] ATBAD with TAAA formation: *n* = 1 [4.3%]	Non applicable
Treatment modality of choice	TEVAR and EVAR: *n* = 2 FEVAR: *n* = 11 BEVAR and EVAR: *n* = 3 Arch BEVAR: *n* = 1 BEVAR: *n* = 2 TEVAR and BEVAR: *n* = 8 T-arch EVAR: *n* = 1 TEVAR, BEVAR, and EVAR: *n* = 2 TEVAR and FEVAR: *n* = 1	TEVAR and BEVAR: *n* = 2 [25%] T-arch-EVAR: *n* = 1 [12.5%] TEVAR and EVAR: *n* = 1 [12.5%] TEVAR, BEVAR and EVAR: *n* = 2 [25%] BEVAR and EVAR: *n* = 1 [12.5%] TEVAR and FEVAR: *n* = 1 [12.5%]	TEVAR and EVAR: *n* = 1 [4.3%] FEVAR: *n* = 11 [47.8%] BEVAR and EVAR: *n* = 2 [8.6%] Arch BEVAR: *n* = 1 [4.3%] BEVAR: *n* = 2 [8.6%] TEVAR and BEVAR: *n* = 6 [26.4%]	Non applicable

**Table 3 medicina-62-00017-t003:** Technical information regarding the placement of the conduit.

Patient Group/Parameter	Conduit [*n* = 8]
Iliac conduit position	Common iliac artery: *n* = 7 [87.5%] Aortic bifurcation: *n* = 1 [12.5%]
Conduit graft	Dacron: *n* = 7 [87.5%] ePTFE ringed: *n* = 1 [12.5%]
Reason for conduit placement	Small iliac artery < 7 mm: *n* = 3 [37.5%] Severely stenotic/occluded iliac artery: *n* = 5 [62.5%]
Further conduit placement	Yes: *n* = 3 [axillary artery: *n* = 2 and subclavian artery: *n* = 1] [37.5%] No: *n* = 5 [62.5%]
Post-op conduit ligation	Yes: *n* = 4 [50%] No: *n* = 4 [50%] [iliofemoral bypass: *n* = 2, aortoiliac bypass: *n* = 1, conduit left inside for second staged intervention: *n* = 1]

**Table 4 medicina-62-00017-t004:** Intra-operative information. Median values are reported for quantitative variables.

Patient Group/Parameter	Total [*n* = 31]	Conduit [*n* = 8]	No Conduit [*n* = 23]	*p*-Value
Type of anesthesia	Epidural: *n* = 1 Local anesthesia and suppression: *n* = 1 General anesthesia: *n* = 30	General anesthesia: *n* = 8 [100%]	General anesthesia: *n* = 22 [95.6%] Spinal: *n* = 1 [4.3%]	Non applicable
Total operative time	243 min	365 min	200 min	0.002
Estimated blood loss [EBL]	820 cc	1190 cc	600 cc	<0.001
Total contrast	340 cc	420 cc	300 cc	0.004

**Table 5 medicina-62-00017-t005:** Post-operative information.

Patient Group/Parameter	Total [*n* = 31]	Conduit [*n* = 8]	No Conduit [*n* = 23]	*p*-Value
Technical success	Yes: 28 No: 3	Yes: *n* = 8 [100%] No: *n* = 0 [0%]	Yes: *n* = 20 [86.9%] No: *n* = 3 [13%]	0.28
Endoleak	Yes: *n* = 6 No: *n* = 25	Yes: *n* = 2 [II EL: *n* = 1 and III EL: *n* = 1 [25%] No: *n* = 6 [75%]	Yes: *n* = 4 [IB EL: *n* = 1 and II EL: *n* = 3 [21.7%] No: *n* = 19 [78.3%]	Non-applicable
Clinical success	Yes: *n* = 25 No: *n* = 6	Yes: *n* = 7 [87.5%] No: *n* = 1 [12.5%]	Yes: *n* = 18 [78.2%] No: *n* = 5 [21.7%]	0.569
ICU stay	Yes: *n* = 20 No: *n* = 11	Yes: *n* = 8 [100%] No: *n* = 0 [0%]	Yes: *n* = 12 [52.1%] No: *n* = 11 [47.9%]	0.015
Median ICU stay	2 d	3 d	2 d	0.817
Post-op complications	Yes: *n* = 23 No: *n* = 8	Yes: *n* = 5 [62.5%] No: *n* = 3 [37.5%]	Yes: *n* = 18 [78.3%] No: *n* = 5 [11.7%]	0.380
Neurological complications	Yes: *n* = 6 No: *n* = 25	Yes: *n* = 2 [25%] [1 pt Horner syndrome, 1 pt hematoma pressure] No: *n* = 6 [75%]	Yes: *n* = 4 [17.4%] No: *n* = 19 [82.6%]	0.639
Respiratory complications	Yes: *n* = 7 No: *n* = 24	Yes: *n* = 4 [50%] No: *n* = 4 [50%]	Yes: *n* = 3 [13%] No: *n* = 20 [87%]	0.031
Bleeding complications necessitating reintervention	Yes: *n* = 2 No: *n* = 29	Yes: *n* = 1 [12.5%] [conduit place bleeding --> SG --> hematoma compression symptoms to nerves --> hematoma drainage] [12.5%] No: *n* = 7 [87.5%]	Yes: *n* = 1 [4.3%] No: *n* = 22 [95.7%]	0.419
AKI	Yes: *n* = 6 No: *n* = 25	Yes: *n* = 1 [RRA occlusion] [12.5%] No: *n* = 7 [87.5%]	Yes: *n* = 5 [21.7%] No: *n* = 18	0.569
Median hospital stay	10 d	12 d	9 d	0.404
Discharged	Alive: 26	Alive: 7 [87.5%]	Alive: 19 [82.6%]	0.746

## Data Availability

Data supporting the reported results are available upon request from the corresponding author.

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
