# Peer review of "Open Iliac Conduits Enabling the New Era of Endovascular Aortic Repair in Hostile Iliofemoral Anatomy: A Single-Center Retrospective Study"

_medicina, 2025, doi:10.3390/medicina62010017_

Round 1

Reviewer 1 Report

Comments and Suggestions for Authors

I would like to start by thanking you for giving me the opportunity to peer review the retrospective study regarding the use of open iliac conduits for endovascular aortic repair in hostile iliofemoral anatomy. The authors of the study showcase that open iliac conduits can achieve great success with minimal complications in severe cases with hostile anatomy as thoracic, branched, or fenestrated endovascular aortic repair. It is a very well-written manuscript regarding a surgical technique that is not yet sufficiently investigated, and which could have a substantial impact on otherwise unoperated patients. In my opinion, some minor revisions could be implemented pending its acceptance for publication.

1. I would suggest making the title more appealing to the readers by reducing its size and adding some extra information regarding the characteristics of the study (prospective or retrospective study, single-center or multicenter). For example, the title could be "Open Iliac Conduits Enabling the New Era of Endovascular Aortic Repair in Hostile Iliofemoral Anatomy: A Single-Center Retrospective Study".

2. The Results section is the Abstract fails to mention p-values, although you have already calculated them. They should be included in the abstract as well.

3. In Figure 1, although you have correctly used arrows to better highlight the anatomical structures, these arrows are not easily distinguished. I would suggest making them bigger and/or adding some different colors.

4. Some of the references date back to as much as 23 years (!) (reference numbers 7,11,14). Replacing them with studies that have been conducted in the last decade would be ideal.

Overall, the authors have effectively gathered and presented the data thoroughly and clearly. A more detailed language revision should be conducted.

Author Response

  1. I would suggest making the title more appealing to the readers by reducing its size and adding some extra information regarding the characteristics of the study (prospective or retrospective study, single-center or multicenter). For example, the title could be "Open Iliac Conduits Enabling the New Era of Endovascular Aortic Repair in Hostile Iliofemoral Anatomy: A Single-Center Retrospective Study".

Response: Thank you for your comment, this has been changed accordingly.

  1. The Results section is the Abstract fails to mention p-values, although you have already calculated them. They should be included in the abstract as well.

Response: P-Values have been added in the Results section of the abstract according to your comment.

  1. In Figure 1, although you have correctly used arrows to better highlight the anatomical structures, these arrows are not easily distinguished. I would suggest making them bigger and/or adding some different colors.

Response: Thank you for your comment. Probably you refer to Figure 2. This has been changed accordingly. Figure legend has been modified as well.

  1. Some of the references date back to as much as 23 years (!) (reference numbers 7,11,14). Replacing them with studies that have been conducted in the last decade would be ideal.

Response: Old references (#7,11,13,14) have been replaced with recent papers, according to your remark (the 4 references that have been added have been published in 2016, 2021, 2023, 2023).

Overall, the authors have effectively gathered and presented the data thoroughly and clearly. A more detailed language revision should be conducted.

Response: Language was reviewed throughout the text.

Reviewer 2 Report

Comments and Suggestions for Authors

Peer review

The article is a single-center retrospective cohort study evaluating the role of retroperitoneal open iliac conduits to facilitate complex endovascular aortic repair in patients with hostile iliofemoral anatomy.

The authors focus on patients undergoing thoracic, branched, or fenestrated endovascular aortic repair, in which large-bore transfemoral access is limited by hostile iliofemoral anatomy, such as small-caliber, calcified, stenotic, or tortuous iliac arteries.

The design is correctly described as a retrospective observational cohort, and the comparison between ROIC and no-conduit patients is restricted to complex endovascular repairs, thereby improving internal comparability.

The introduction provides a reasonable clinical background and justification for the study question, but it does not fully cover or critically frame all relevant literature for a higher‑tier journal.

The discussion of prior ROIC series is very superficial; there is no concise summary of existing outcomes (morbidity, mortality, ischemic complications) to show exactly what is new beyond “rarely isolated.

The results are generally understandable and supported by appropriate tables, but the presentation would benefit from sharper focusing on a few key endpoints and cleaner separation of total versus subgroup data.

The novelty and originality are modest and mainly incremental.

The main conclusion is that planned retroperitoneal open iliac conduits are a safe and useful enabling technique for complex endovascular aortic repair in patients with hostile iliofemoral anatomy, achieving high technical and clinical success without added early ischemic complications, despite longer and more resource-intensive procedures.

The conduit group (n=8) is tiny, so any formal significance testing (p values) for between‑group differences is statistically fragile and at high risk of type I and type II error. The follow-up is limited to 12 months, and the outcomes are relatively rare, making time-to-event methods underpowered; presenting crude proportions without overinterpreting differences would be more defensible.

The novelty is modest and mainly lies in providing contemporary, conduit-focused data within a complex EVAR cohort, rather than in introducing a new technique.

The key comparative inference (ROIC vs no-conduit) is based on only 8 conduit patients and 23 controls, which is underpowered given the number of variables and endpoints examined.

The paper largely confirms that ROIC is feasible and safe as an enabling access technique, a concept well established in prior literature; the added value of a small single-center series without long follow-up or advanced analysis is modest.

There is no comparison with open surgery, which would have sharpened the claimed “indispensable niche” and increased clinical impact.

Author Response

The article is a single-center retrospective cohort study evaluating the role of retroperitoneal open iliac conduits to facilitate complex endovascular aortic repair in patients with hostile iliofemoral anatomy.

The authors focus on patients undergoing thoracic, branched, or fenestrated endovascular aortic repair, in which large-bore transfemoral access is limited by hostile iliofemoral anatomy, such as small-caliber, calcified, stenotic, or tortuous iliac arteries.

The design is correctly described as a retrospective observational cohort, and the comparison between ROIC and no-conduit patients is restricted to complex endovascular repairs, thereby improving internal comparability.

Comment 1: The introduction provides a reasonable clinical background and justification for the study question, but it does not fully cover or critically frame all relevant literature for a higher‑tier journal.

Response: The Introduction has been expanded to provide more information on the subject. The following text has been added "According to previous research, retroperitoneal iliac exposure during endovascular repair of complex aortic aneurysms has been associated with lower technical success rates, albeit without increasing intraoperative access complications or major adverse events [7]. Specifically, an 28% major adverse events rate was recorded in the conduit cohort, with no statistically significant difference from the non-conduit, while technical success rate was 83% among patients in the conduit compared with 98% in the standard group [7]. Others summarized available evidence conducting a systematic review and reported a pooled estimate for overall technical success >95%, on top of a non-negligible rate of bleeding (10%), wound (5%) and any peri-procedural complications (32%), among patients requiring an iliac conduit.[9] Notably, this metanalysis reported a statistically significant almost 3-fold odds ratio for death and a 2.38 odds ratio for bleeding complications among patients requiring a conduit. Contemporary series rarely isolate ROIC outcomes in elective t/b/fEVAR, nor compare them to no-conduit cohorts under modern reporting standards and comparative analyses are still needed to determine the best strategy to address challenging iliac artery accesses". This is found in Page 2, Lines 62-76.

Comment 2: The discussion of prior ROIC series is very superficial; there is no concise summary of existing outcomes (morbidity, mortality, ischemic complications) to show exactly what is new beyond “rarely isolated.

Response: Relevant information has been added in the Introduction. Please, see previous answer in the comment as well.

Comment 3: The results are generally understandable and supported by appropriate tables, but the presentation would benefit from sharper focusing on a few key endpoints and cleaner separation of total versus subgroup data.

Response: The structure of the Results section has been modified and the separation between primary and secondary outcomes has been made clearer, with primary endpoints reported in a separate paragraph to give more emphasis in those key findings (paragraph #3.3, Page 7, Lines 217-247). Text that has been added reads as follows "3.3 Primary Outcomes, 3.3.1. Technical success: Technical success was achieved in all patients in the conduit group and in 20/23 patients in the non-conduit group, as per Oderich reporting standards. This difference was not statistically significant (p=0.28). Three patients in the non-conduit group had intraoperative rupture of the aneurysm, requiring massive blood transfusion, resulting in complications as disseminated intravascular coagulation (DIC), transfusion-related acute lung injury (TRALI) and finally in death within 24h of the operation.

3.3.2 Clinical success: Clinical success was observed in all  but one patient in the conduit group and in 18/23 patients in the non-conduit group, also without evidence of statistically significant difference (p=0.569). One patient in the conduit-group suffered from iliac limb thrombosis leading to non-reversible ischemia, thus resulting in an above-the-knee amputation. Among the patients in the non-conduit group, three of them did not achieve technical success, thus excluding them by definition from achieving clinical success. Inbetween the rest two patients, one had the branch to the left renal artery thrombosed and the second one had type IB endoleak in the 30-d follow-up CTA, necessitating in both cases reintervention, either with AngioJet thrombectomy and renal artery stent relining or with iliac limb stent relining, respectively.

3.3.3. In-hospital mortality: In-hospital mortality occurred in 1/8 patients in the conduit and in 4/23 patients in the non-conduit group. The patient in the conduit-group deceased during her 100th post-operative day, after a burdensome hospitalization complicated with iliac limb thrombosis requiring femoral amputation, intestinal ischemia requiring multiple enterectomy operations from general surgeons, two episodes of cardiac arrest and finally an upper respiratory infection which finally resulted in patients’ death. In the non-conduits group, three patients died within 24h of the operation, as previously stated in section 3.3.1, while one patient died during his 83rd post-operative day, after a prolonged ICU stay, due to severe, recurring upper respiratory infection with pandrug-resistant (PDR) bacteria".

Moreover, in Methods, Page 3, Lines 115-120 it has been delinated that primary and secondary ouitcomes will be analyzed.

Finally, a sentence has been added to clarify the fact that statistical comparisons regard patients undergoing Fenestrated or Branched EVAR with or without conduits (Page 4, Lines 139,140). 

Comment 4: The novelty and originality are modest and mainly incremental.

Response: We could not disagree with this argument. The current analysis presents contemporary outcomes of complex endovascular aneurysm repair in patients with narrow access vessels with the use of iliac conduits which indeed represent an established surgical technique. Our results should be viewed in this context. Therefore, we have expanded the limitations section in order to underline these constraints. Specifically in Page 14, Lines 419-423, teh follwing has been added "Moreover, taking into account that the focus of the current report is to provide contemporary results of complex EVAR with the use of open surgical conduits to overcome narrow access, rather than introducing a new technique, novelty of the present findings may be modest".

Comment 5: The main conclusion is that planned retroperitoneal open iliac conduits are a safe and useful enabling technique for complex endovascular aortic repair in patients with hostile iliofemoral anatomy, achieving high technical and clinical success without added early ischemic complications, despite longer and more resource-intensive procedures.

Response: Thank you for this remark. We have changed th Conclusion accordingly which now reads as "Our single-center experience confirms that retroperitoneal open iliac conduits remain an indispensable tool in contemporary practice. When severe calcification, occlusive disease, extreme tortuosity, or small-caliber external iliac arteries preclude safe transfemoral delivery of next-generation endografts, the deliberate use of a surgical conduit converts anatomical exclusion criteria into technical feasibility. Although placement of an open conduit is associated with longer operative times, greater blood loss, and more intensive postoperative care, it offers a controlled, durable access solution that enables the full spectrum of modern branched, fenestrated, and total-arch endovascular reconstructions. In total, planned retroperitoneal open iliac conduits are a safe and useful enabling technique for complex endovascular aortic repair in patients with hostile iliofemoral anatomy, achieving high technical and clinical success without added early ischemic complications, despite longer and more resource-intensive procedures". This can be seen in Page 14, Lines 438-450.

Comment 6: The conduit group (n=8) is tiny, so any formal significance testing (p values) for between‑group differences is statistically fragile and at high risk of type I and type II error. The follow-up is limited to 12 months, and the outcomes are relatively rare, making time-to-event methods underpowered; presenting crude proportions without overinterpreting differences would be more defensible.

Response: Although we agree with this remark, we chose to keep the statistical analysis/statistical comparisons, rather than make a descriptive presentation of the outcomes. We have included a relevant comment in the limitation section to underline the possibility for a Type I or II error due to this small sample size. In Page 14, Lines 416-419, teh following text has been added "Regarding the small sample size, one could argue against performing a formal statistical comparison between these small patient groups. We chose to perform a statistical comparison, recognizing the high potential for a Type I and II statistical error, instead of just making a descriptive report of the outcomes".

Comment 7: The novelty is modest and mainly lies in providing contemporary, conduit-focused data within a complex EVAR cohort, rather than in introducing a new technique. The key comparative inference (ROIC vs no-conduit) is based on only 8 conduit patients and 23 controls, which is underpowered given the number of variables and endpoints examined.

Response: Again, please see response of previous comments, since a relevant limitation has already been included.

The paper largely confirms that ROIC is feasible and safe as an enabling access technique, a concept well established in prior literature; the added value of a small single-center series without long follow-up or advanced analysis is modest.

Response: Again, please see response of previous comments.

Comment 8: There is no comparison with open surgery, which would have sharpened the claimed “indispensable niche” and increased clinical impact.

Response: We thank you for this thoughtful comment. We fully agree that open surgical repair remains an important treatment option for complex aortic aneurysms. However, we believe that a direct comparison between open repair and the present cohort would not be methodologically appropriate. Patients selected for fenestrated and/or branched endovascular repair via open iliac conduits represent a distinct subgroup: they typically have extensive aneurysm disease, severe access vessel limitations, and comorbidity profiles that make them unsuitable candidates for open surgery.

Given these differences in baseline characteristics, introducing a comparison with open repair would risk implying clinical equivalence between groups that are inherently non-comparable and subject to substantial selection bias. For this reason, the current study focuses on the role and outcomes of open iliac conduits within the context of endovascular management, where they serve as an adjunct enabling F/BEVAR in anatomically challenging patients.

We have included a relevant comment in the limitations section of the manuscript, emphasizing that open iliac conduits serve a specific, indispensable niche within the endovascular treatment paradigm, rather than as an alternative to open repair. In Page 14, Lines 427-436, the following text has been added "Finaly, the current study did not consider patients undergoing open surgical repair of complex aortic aneurysms. Patients selected for fenestrated and/or branched endovascular repair via open iliac conduits represent a distinct group than those selected for open surgery, typically presenting more extensive aortic disease and comorbidity profiles that make them poor candidates for open surgery. Accordingly, introducing a comparison with open repair did not seem methodologically appropriate taking into account that these groups are inherently non-comparable and subject to substantial selection bias. For this reason, the current study focuses on the role and outcomes of open iliac conduits within the context of endovascular management, where they serve as an adjunct to facilitate implantation of endografts in patients with hostile iliac anatomy".